# Diagnostic Approach to Traumatic Axonal Injury of the Spinothalamic Tract in Individual Patients with Mild Traumatic Brain Injury

**DOI:** 10.3390/diagnostics9040199

**Published:** 2019-11-21

**Authors:** Sung Ho Jang, Han Do Lee

**Affiliations:** 1Department of Physical Medicine and Rehabilitation, College of Medicine, Yeungnam University, HyunChungro 170, Daegu 705-717, Korea; rehab6467@hanmail.net; 2Department of Physical Therapy, College of Natural Science, Ulsan College University, Bongsuro 101 Dongku, Ulsan 44022, Korea

**Keywords:** Spinothalamic tract, Mild traumatic brain injury, Traumatic axonal injury, Diffusion tensor tractography

## Abstract

Objectives: We investigated an approach for the diagnosis of traumatic axonal injury (TAI) of the spinothalamic tract (STT) that was based on diffusion tensor tractography (DTT) results and a statistical comparison of individual patients who showed central pain following mild traumatic brain injury (mTBI) with the control group. Methods: Five right-handed female patients in their forties and with central pain following mTBI and 12 age-, sex-, and handedness-matched healthy control subjects were recruited. After DTT reconstruction of the STT, we analyzed the STT in terms of three DTT parameters (fractional anisotropy (FA), mean diffusivity (MD), and fiber number (FN)) and its configuration (narrowing and tearing). To assess narrowing, we determined the area of the STT on an axial slice of the subcortical white matter. Results: the FN values were significantly lower in at least one hemisphere of each patient when compared to those of the control subjects (*p* < 0.05). Significant decrements from the STT area in the control group were observed in at least one hemisphere of each patient (*p* < 0.05). Regarding configurational analysis, the STT showed narrowing and/or partial tearing in at least one hemisphere of each of the five patients. Conclusions: Herein, we demonstrate a DTT-based approach for the diagnosis of TAI of the STT. The approach involves a statistical comparison between DTT parameters of individual patients who show central pain following mTBI and those of an age-, gender-, and handedness-matched control group. We think that the method described in this study can be useful in the diagnosis of TAI of the STT in individual mTBI patients.

## 1. Introduction

A traumatic brain injury (TBI) might be classified as mild, moderate, or severe. Mild TBI (mTBI), which accounts for 70–90% of all TBI, usually shows negative results on conventional brain magnetic resonance imaging (MRI) [1,2]. Since the 1960s, autopsy-based pathological studies have investigated traumatic axonal injuries (TAIs) and described them as resulting from tearing of axons due to the shearing forces associated with acceleration, deceleration, and rotation of the brain during mTBI [3,4,5]. TAI is a brain injury in which scattered lesions in white matter tracts, as well as gray matter, occur over a widespread area. However, because conventional brain MRI is insufficiently sensitive to detect TAI in mTBI, a diagnosis of TAI in living patients with mTBI was impossible for a long time [6,7,8]. Hundreds of studies have used DTI to demonstrate TAI in mTBI patients since the development of diffusion tensor imaging (DTI) in the 1990s [6,7,8]. However, most of these studies have focused on demonstrating TAI in an mTBI group comprised of a number of patients. Nonetheless, the detection of a TAI in an individual patient is important for both clinical management and prognosis prediction in the clinical field.

The DTI method using ROIs can yield false results due to the high variability among individuals in the anatomical location of a neural tract, and it has lower reliability than the diffusion tensor tractotography (DTT) method [7,8,9,10]. The main advantage of DTT over DTI is that it allows for the entire neural tract to be evaluated by measuring DTT parameters [7,8]. In addition, configurational analysis of the reconstructed neural tracts can indicate abnormalities, such as tearing, narrowing, or discontinuations, which have been used to detect TAI of neural tracts in mTBI [8,11]. Furthermore, the DTT method is reported to have excellent reliability, as well as greater repeatability, than the DTI method [10]. As a result, the DTT method appears to be more effective than the DTI method when attempting to detect TAI in an individual patient [8]. However, methods for detecting the TAI of neural tracts in mTBI have not been fully established, although a few methods, such as DTT parameter measurement, configurational analysis, and DTI parameter measurement using ROIs, have been suggested [7,8,11,12,13,14].

In this study, we investigated a diagnostic approach for the diagnosis of TAI of the spinothalamic tract (STT), an injury that produces central pain. The diagnostic method is based on a statistical comparison of selected DTT parameters of an individual patient who has central pain following mTBI with those of an age-, gender-, and handedness-matched control group [11,13,15].

## 2. Case Report

Five right-handed female patients (aged 42–48 years, mean age 45.00 ± 3.00 years) with mTBI and 12 right-handed normal control female subjects (aged 41–49 years, mean age 45.71 ± 4.15 years) were included in this study and none of them had a history of neurological, physical, or psychiatric illness. Patients were recruited according to the following inclusion criteria: (1) loss of consciousness for <30 min, post-traumatic amnesia for ≤24 h, and an initial Glasgow Coma Scale score of 13–15; (2) presence of central pain characteristic of neuropathic pain, stimulation-independent pain (shooting, lancinating, burning, electric shock-like sensation, or paresthesia (crawling, itching, or tingling sensation)), or stimulus-evoked pain (hyperalgesia or allodynia by environmental stimuli) [16,17,18,19,20]; (3) no specific lesion being observed on brain MRI (T1-weighted, T2-weighted, and fluid-attenuated inversion recovery images); (4) age at the time of head trauma, 40–49 years; (5) no radiculopathy or peripheral neuropathy on electromyography and nerve conduction study; and, (6) no musculoskeletal problem (e.g., myofascial pain syndrome, complex regional pain syndrome, or heterotopic ossification). Table 1 summarizes the demographic and clinical data for the five patients and 12 control subjects. All of the patients and control subjects provided signed, informed consent, and our institutional review board (approval number of Yeungnam University Hospital institutional review: YUMC-2018-09-007) approved the study protocol.

DTI data were obtained while using a six-channel head coil on a 1.5T MRI scanner (Gyroscan Intera; Philips Medical System, Best, Netherlands) with single-shot echo-planar imaging at an average of 3.44 ± 4.63 months after the onset of TBI. We acquired 70 contiguous slices parallel to the anterior commissure-posterior commissure line for each of the 32 non-collinear diffusion sensitizing gradients. The DTI parameters were, as follows: acquisition matrix = 96 × 96; reconstructed to matrix = 192 × 192; field of view = 240 mm × 240 mm; repetition time = 10,398 ms; echo time = 72 ms; parallel imaging reduction factor = 2; echo-planar imaging factor = 59; b = 1000 s/mm^2^; number of excitations = 1; and, slice thickness = 2.5 mm. Prior to fiber tracking, eddy current correction was applied to correct for head motion effects and image distortion by using the Functional Magnetic Resonance Imaging of the Brain (FMRIB) Software Library; the default tractography option in the FMRIB library (5000 streamline samples, 0.5 mm step lengths, curvature thresholds = 0.2) was used for fiber tracking [21,22]. 5000 streamline samples were calculated and generated from the seed ROI while using this fiber-tracking method, reflecting both the dominant and non-dominant diffusion orientations in each voxel to reveal brain region connections. For the reconstruction of the STT, the seed ROI was located at an isolated STT area (posterolateral to the inferior olivary nucleus and anterior to the inferior cerebellar peduncle in the medulla) and two target ROIs were placed on the portion of the ventro-postero-lateral nucleus of the thalamus and on the primary somatosensory cortex on axial images [23]. A threshold of two streamlines was applied to the fiber-tracking results. The fractional anisotropy (FA) and mean diffusivity (MD) values, as well as the fiber number (FN), for the STT, were obtained in both hemispheres. We defined partial tearing as a partial or isolated defect in the reconstructed STT for the configurational analysis. For narrowing assessment, we measured the STT area on an axial slice of the subcortical white matter by measuring the length and breadth of the pixels (1.25 mm). 

## 3. Statistical Analysis

Statistical analyses were performed while using SPSS software (v. 25.0; SPSS, Chicago, IL, USA). We performed analysis using Bayesian statistics for the determination of differences in FA, MD, FN, and STT area of each patient and the respective mean values of the control group [24].

## 4. Results

Table 2 summarizes the results of the Bayesian statistical analyses comparing the DTT parameters and the STT areas of each individual patient with the mean values of the control group. Significant differences were not observed for the FA and MD values of both hemispheres in each of the five patients when compared with the mean values for the control subjects (*p* > 0.05). However, compared to the control subjects, the FN values were significantly lower in one hemisphere in three individual patients (patients 1 and 2, right hemisphere; patient 3, left hemisphere) and in both hemispheres of patients 4 and 5 (*p* < 0.05; Table 2). In addition, the STT areas were significantly lower in one hemisphere in two patients (patients 1 and 2, right hemisphere only) and in both hemispheres of patients 3, 4, and 5 when compared to the control group (*p* < 0.05). With regard to configurational analysis, the STT showed narrowing and/or partial tearing in at least one hemisphere of each of the five patients (Figure 1). 

## 5. Discussion

In this study, we recruited individual patients who exhibited clinical results that were compatible with those associated with a diagnosis of TAI of the STT. Excluding the DTT findings, those conditions, which have been demonstrated in previous studies, are as follows: (1) head trauma history that is compatible with mTBI, (2) development of neuropathic pain after the head trauma, and (3) absence of peripheral nerve injury and musculoskeletal problems [1,8,11]. In this study, we investigated a method for the diagnosis of TAI of the STT in individual patients. The method compared the DTT parameters and configurational analysis results for the STT of individual patients with those of a group of control subjects. The comparison results indicated the following: (1) the FN of the STT in at least one hemisphere was significantly decreased in each of the five individual patients, but there were no significant differences in the FA and MD values; and, (2) in at least one hemisphere of each of the five patients the STT showed narrowing and/or partial tearing. The FA value represents the state of white matter organization by indicating the degree of directionality, while the MD value indicates the magnitude of water diffusion [25,26]. The FN value indicates the number of voxels that were included in a neural tract, thereby suggesting the total number of fibers within that tract [25,26]. Therefore, a low FN value for the STT can indicate an injury of that STT, regardless of the lack of evidence of changes in the FA and MD results [25,26]. The configurational analysis of the STT showed narrowing and/or partial tearing in one or both hemispheres in all five patients. In addition, narrowing was demonstrated by measuring the area of the STT in the subcortical white matter region. The FN and configurational results suggest that the STTs were injured in at least one hemisphere in all five patients. The STT injuries identified in this study appear to indicate the presence of TAI because conventional brain MRI of those patients showed no new brain lesions [2,7,8,27]. Therefore, decrement of FN in patients compared with the controls due to TAI in patients with mild TBI.

To date, three methods for the diagnosis of TAI in mTBI have been suggested: (1) configurational analysis of a DTT-reconstructed neural tract; (2) the measurement of DTI parameters while using ROIs that are applied on the partial injury site of a neural tract on DTT; and, (3) statistical comparison of DTT parameters of a neural tract of an individual patient with those for control subjects [8,11,12,14]. Most of the studies that have detected TAI of neural tracts in mTBI have employed configurational analysis of DTT-reconstructed neural tracts; in those studies, the abnormal configurational results were classified as discontinuation, narrowing, partial tearing, non-reconstruction, or decreased neural connectivity of the neural tract [11,12,14,15]. A previous study measured DTI parameters in ROIs in the partially injured areas of a three-dimensionally DTT-reconstructed optic radiation in a patient with visual field defect following mTBI and compared the results with those for the control subjects [14]. Another study reported on a Bayesian statistics-based comparison of DTT parameters of a neural tract between an individual patient who displayed apathy following mTBI and a group of sex-matched control subjects [12]; the results demonstrated injury of the prefronto-caudate tract in the patient [12]. However, in that study, handedness was not considered and age was not thoroughly matched. By contrast, we recruited the age-, sex-, and handedness-matched control subjects in accordance with the recommendation of Shenton et al., although we employed the same statistical method (Bayesian statistics) in the present study [13]. In addition, to confirm the presence of narrowing in our configurational analysis, we measured the STT areas of the patients and the control subjects and confirmed the presence of significant differences by using Bayesian-based statistical analysis.

In conclusion, herein, we demonstrate an approach to the diagnosis of TAI of the STT that uses a statistical comparison of the DTT parameters of an individual patient experiencing central pain following mTBI with those of an age-, gender-, and handedness-matched normal control group. We think that the method can be effective when diagnosing TAI of the STT in mTBI patients. In the past, brain MRI was used to diagnose TAI, which is a disadvantage in that MRI lesions are not diagnosed. However, DTI has a big advantage in detecting TAI in the existing MRI methods. Further studies are needed for determining the effectiveness of this method in the diagnosis of TAI in other neural tracts. However, there are limitations of DTT that should be considered when interpreting our results [28,29,30]. First, although DTT is a powerful anatomic imaging tool that can demonstrate gross fiber architecture, it can be difficult for DTT to reflect all fibers, particularly small fibers; thus, DTT might underestimate or overestimate fiber tract configuration. Second, brain regions with fiber complexity and/or fiber crossing can prevent DTT from accurately reflecting the underlying fiber architecture.

## Figures and Tables

**Figure 1 diagnostics-09-00199-f001:**
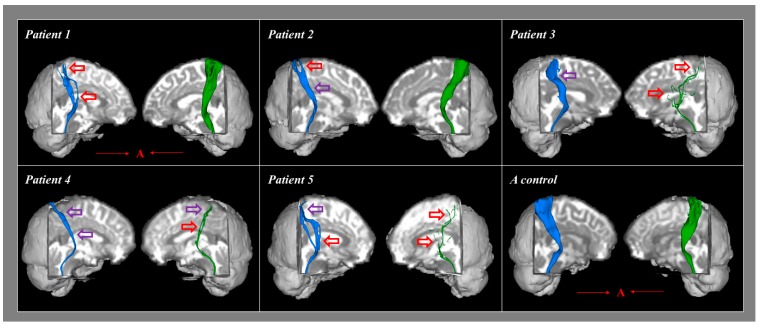
Results of diffusion tensor tractography (DTT) for the spinothalamic tract of five patients and a representative control group subject. Partially torn and narrowed areas are marked with red arrows while narrowed areas are marked with purple arrows (blue color is right spinothalamic tract and green color is left spinothalamic tract). A (red): anterior direction.

**Table 1 diagnostics-09-00199-t001:** Demographic and clinical data for the patients and control subjects.

	Patient 1	Patient 2	Patient 3	Patient 4	Patient 5	Controls (*n* = 12)
Age (year)	42	48	42	48	45	45.71 ± 4.15
Duration to DTT	2 days	14 months	1 month	2 months	1 month	-
Central pain site (VAS)	Head (5), right arm, and leg (5)	Left head, arm, and leg (5)	Head (4), right leg (4)	Head (2), both legs (3)	Head (5), both arms and legs (4)	No pain
Characteristics of central pain	Tingling sensation & allodynia	Tingling sensation & allodynia	Electric shock-like sensation & allodynia	Tingling & electric shock-like sensations	Tingling sensation & allodynia	No pain

DTT: diffusion tensor tractography. VAS: visual analog scale score.

**Table 2 diagnostics-09-00199-t002:** Results of Bayesian statistics analyses of diffusion tensor tractography parameters and spinothalamic tract (STT) areas of the individual patients and the group of control subjects.

			Patient 1	Patient 2	Patient 3	Patient 4	Patient 5	Controls
Diffusion Tensor Tractography Parameters								
[Significance] *^a^*	FA	Right	0.44 [0.32]	0.41 [0.38]	0.40 [0.29]	0.42 [0.48]	0.45 [0.24]	0.42 ± 0.03
		Left	0.41 [0.31]	0.39 [0.15]	0.45 [0.26]	0.47 [0.12]	0.43 [0.46]	0.42 ± 0.03
	MD	Right	0.84 [0.31]	0.81 [0.21]	0.86 [0.39]	0.79 [0.15]	0.90 [0.42]	0.88 ± 0.08
		Left	0.89 [0.21]	0.80 [0.35]	0.77 [0.22]	0.78 [0.26]	0.76 [0.19]	0.82 ± 0.07
	FN	Right	298 [0.02] *^c^*	660 [0.04] *^c^*	1935 [0.23]	438 [0.02] *^c^*	632 [0.03] *^c^*	1576.81 ± 461.74
		Left	1824 [0.41]	1934 [0.36]	38 [0.01] *^c^*	202 [0.02] *^c^*	20 [0.01] *^c^*	1680.90 ± 656.56
Estimated effect size *^b^* (95% CI)	FA	Right	0.501 (−1.913, 2.916)	−0.327 (−2.757, 2.101)	−0.598 (−3.002, 1.805)	−0.050 (−2.491, 2.389)	0.765 (−1.614, 3.146)	
		Left	−0.539 (−2.950, 1.871)	−1.115 (−3.425, 1.195)	0.703 (−1.686, 3.093)	1.303 (−1.016, 3.532)	0.086 (−2.353, 2.526)	
	MD	Right	−0.533 (−2.945, 1.877)	−0.888 (−3.247, 1.470)-	−0.286 (−2.718, 2.146)	−1.110 (−3.422, 1.201)	0.217 (−2.218, 2.653)	
		Left	0.881 (−1.478, 3.242)	0.412 (−2.836, 2.010)	1.063 (−3.202, 1.536)	−0.676 (−3.087, 1.694)	−0.965 (−3.309, 1.378)	
	FN	Right	−2.001 (−3.993, −0.007)	−1.784 (−3.876, 0.307)	0.792 (−1.584, 3.168)	−2.072 (−4.028, −0.116)	−1.824 (−3.899, 0.250)	
		Left	0.228 (−2.207, 2.663)	0.401 (−2.022, 2.825)	−2.091 (−4.037, −0.146)	−1.951 (−3.968, 0.647)	−2.106 (−4.044, −0.168)	
Spinothalamic Tract Area								
STT area (mm) [Significance] *^a^*		Right	12.50 [0.02] *^c^*	9.37 [0.02] *^c^*	18.75 [0.03] *^c^*	11.50 [0.02] *^c^*	16.87 [0.03] *^c^*	98.95 ± 17.86
		Left	101.25 [0.58]	78.12 [0.34]	1.56 [0.04] *^c^*	3.12 [0.04] *^c^*	1.56 [0.04] *^c^*	91.40 ± 9.66
Estimated effect size *^b^* (95% CI)		Right	−2.03 (−3.97, −0.10)	−2.07 (−3.96, −0.17)	−1.96 (−4.00, 0.06)	−2.00 (−3.98, −0.020)	−1.92 (−4.01, 0.15)	
		Left	1.02 (−1.78, 3.82)	−0.10 (−3.14, 2.93)	−1.52 (−4.00, 0.96)	−1.50 (−4.00, 1.00)	−1.52 (−4.00, 0.96)	

DTT: diffusion tensor tractography, FA: fractional anisotropy, MD: mean diffusivity, FN: fiber number, STT: spinothalamic tract. [Significance]: *^a^* Bayesian hypothesis 1-tailed test for probability that a member of the control population has a lower FA or FN score, a higher MD score, or a lower STT cross-sectional area than each patient. *^b^* Effective size (Zcc) for difference between the patient and the control subject group. *^c^ p* < 0.05.

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
