# Peer review of "Diagnostic Approach to Traumatic Axonal Injury of the Spinothalamic Tract in Individual Patients with Mild Traumatic Brain Injury"

_diagnostics, 2019, doi:10.3390/diagnostics9040199_

Round 1

Reviewer 1 Report

This study reconstructed spinothalamic tract and statistically analyzed its number in mTBI patients compared with that in control subjects. The study design is solid and easy to understand. However, the authors need to clarify that why the spinothalamic tract is interesting and why there is a decrease in fiber numbers in patients compared with the controls.

Author Response

Decrement of FN in patients compared with the controls due to TAI in patients with mild TBI.

Therefore, we revised in the manuscript.

Reviewer 2 Report

The authors describe the use of neuroimaging of tracts (DTI and DTT) in detecting spinothalamic axonal injury following mild head injury. The methods and analyses are appropriate. The discussion is measured and mentions caveats to the technique. I think this is an excellent paper of clinical importance and of interest to physicians in pain medicine, neurosurgery and neurology.

Author Response

Thank you for your comment.

Reviewer 3 Report

The authors investigated the novel approach for the diagnosis of traumatic axonal injury (TAI) of the spinothalamic tract (STT) based on diffusion tensor tractography (DTT). And, they also performed the statistical comparison between individual patients who showed central pain following mild traumatic brain injury (mTBI) and the control group. The results well represent the validity of the diagnostic DTT-based approach for TAI and STT. And, it is well emphasized the useful for the diagnosis of STT TAI through the MRI imaging and statistical comparisons of the age, gender, and handers of individual patients with post-mTBI central pain. However, there are some parts that need minor revisions. 1. There is no content available for readers to understand TBI and TAI diseases. 2. Methodology is introduced in Introduction. 3. There is no discussion about the clinical implications by the improvement of old fashions when using the new diagnostic method suggested by the authors.

Author Response

Thank you for your comment. So, we revised as follow.

Round 2

Reviewer 1 Report

The authors have address all my question. I have no other comments.